# Stochastic Gradient Methods for Distributionally Robust Optimization with $f$-divergences

**Hongseok Namkoong**
Stanford University
hnamk@stanford.edu

**John C. Duchi**
Stanford University
jduchi@stanford.edu

## Abstract

We develop efficient solution methods for a robust empirical risk minimization problem designed to give calibrated confidence intervals on performance and provide optimal tradeoffs between bias and variance. Our methods apply to distributionally robust optimization problems proposed by Ben-Tal et al., which put more weight on observations inducing high loss via a worst-case approach over a non-parametric uncertainty set on the underlying data distribution. Our algorithm solves the resulting minimax problems with nearly the same computational cost of stochastic gradient descent through the use of several carefully designed data structures. For a sample of size $n$, the per-iteration cost of our method scales as $O(\log n)$, which allows us to give optimality certificates that distributionally robust optimization provides at little extra cost compared to empirical risk minimization and stochastic gradient methods.

## 1   Introduction

In statistical learning or other data-based decision-making problems, it is desirable to give solutions that come with guarantees on performance, at least to some specified confidence level. For tasks such as driving or medical diagnosis where safety and reliability are crucial, confidence levels have additional importance. Classical techniques in machine learning and statistics, including regularization, stability, concentration inequalities, and generalization guarantees [6, 25] provide such guarantees, though often a more fine-tuned certificate—one with *calibrated* confidence—is desirable. In this paper, we leverage techniques from the robust optimization literature [e.g. 2], building an uncertainty set around the empirical distribution of the data and studying worst case performance in this uncertainty set. Recent work [15, 13] shows how this approach can give (i) calibrated statistical optimality certificates for stochastic optimization problems, (ii) performs a natural type of regularization based on the variance of the objective and (iii) achieves fast rates of convergence under more general conditions than empirical risk minimization by trading off bias (approximation error) and variance (estimation error) optimally. In this paper, we propose efficient algorithms for such distributionally robust optimization problems.

We now provide our formal setting. Let $\mathcal{X} \subset \mathbb{R}^d$ be a compact convex set, and for a convex function $f : \mathbb{R}_+ \to \mathbb{R}$ with $f(1) = 0$, define the $f$-divergence between distributions $P$ and $Q$ by $D_f(P\|Q) = \int f(\frac{dP}{dQ})dQ$. Letting $\mathcal{P}_{\rho,n} := \{p \in \mathbb{R}^n : p^\top \mathbb{1} = 1, p \geq 0, D_f(p\|\mathbb{1}/n) \leq \frac{\rho}{n}\}$ be an *uncertainty set* around the uniform distribution $\mathbb{1}/n$, we develop methods for solving the robust empirical risk minimization problem

$$\underset{x \in \mathcal{X}}{\text{minimize}} \quad \sup_{p \in \mathcal{P}_{\rho,n}} \sum_{i=1}^n p_i \ell_i(x). \tag{1}$$

In problem (1), the functions $\ell_i : \mathcal{X} \to \mathbb{R}_+$ are convex and subdifferentiable, and we consider the situation in which $\ell_i(x) = \ell(x; \xi_i)$ for $\xi_i \overset{\text{iid}}{\sim} P_0$. We let $\ell(x) = [\ell_1(x) \cdots \ell_n(x)]^\top \in \mathbb{R}^n$ denote the vector of convex losses, so the robust objective (1) is $\sup_{p \in \mathcal{P}_{\rho,n}} p^T \ell(x)$.

A number of authors show how the robust formulation (1) provides guarantees. Duchi et al. [15] show that the objective (1) is a convex approximation to regularizing the empirical risk by variance,

$$\sup_{p \in \mathcal{P}_{\rho,n}} \sum_{i=1}^n p_i \ell_i(x) = \frac{1}{n} \sum_{i=1}^n \ell_i(x) + \sqrt{\frac{\rho}{n} \mathrm{Var}_{P_0}(\ell(x; \xi))} + o_{P_0}(n^{-\frac{1}{2}}) \tag{2}$$

uniformly in $x \in \mathcal{X}$. Since the right hand side naturally trades off good loss performance (approximation error) and minimizing variance (estimation error) which is usually non-convex, the robust formulation (1) provides a convex regularization for the standard empirical risk minimization (ERM) problem. This trading between bias and variance leads to certificates on the optimal value $\inf_{x \in \mathcal{X}} \mathbb{E}_{P_0}[\ell(x; \xi)]$ so that under suitable conditions, we have

$$\lim_{n \to \infty} \mathbb{P}\left(\inf_{x \in \mathcal{X}} \mathbb{E}_{P_0}[\ell(x; \xi)] \leq u_n\right) = \mathbb{P}\left(W \geq -\sqrt{\rho}\right) \quad \text{for } W \sim \mathsf{N}(0, 1) \tag{3}$$

where $u_n = \inf_{x \in \mathcal{X}} \sup_{p \in \mathcal{P}_{\rho,n}} p^T \ell(x)$ is the optimal robust objective. Duchi and Namkoong [13] provide finite sample guarantees for the special case that $f(t) = \frac{1}{2}(t-1)^2$, making the expansion (2) more explicit and providing a number of consequences for estimation and optimization based on this expansion (including fast rates for risk minimization). A special case of their results [13, §3.1] is as follows. Let $\widehat{x}^{\mathrm{rob}} \in \mathrm{argmin}_{x \in \mathcal{X}} \sup_{p \in \mathcal{P}_{\rho,n}} p^T \ell(x)$, let $\mathsf{VC}(\mathcal{F})$ denote the VC-(subgraph)-dimension of the class of functions $\mathcal{F} := \{\ell(x; \cdot) \mid x \in \mathcal{X}\}$, assume that $M \geq \ell(x; \xi)$ for all $x \in \mathcal{X}, \xi \in \Xi$, and for some fixed $\delta > 0$, define $\rho = \log \frac{1}{\delta} + 10\, \mathsf{VC}(\mathcal{F}) \log \mathsf{VC}(\mathcal{F})$. Then, with probability at least $1 - \delta$,

$$\mathbb{E}_{P_0}[\ell(\widehat{x}^{\mathrm{rob}}; \xi)] \leq u_n + O(1)\frac{M\rho}{n} \leq \inf_{x \in \mathcal{X}} \left\{ \mathbb{E}_{P_0}[\ell(x; \xi)] + 2\sqrt{\frac{2\rho \mathrm{Var}_{\widehat{P}_n}(\ell(x; \xi))}{n}} \right\} + O(1)\frac{M\rho}{n} \tag{4}$$

For large $n$, evaluating the objective (1) may be expensive; with fixed $p = \mathbb{1}/n$, this has motivated an extensive literature in stochastic and online optimization [27, 23, 19, 16, 18]. The problem (1) does not admit quite such a straightforward approach. A first idea, common in the robust optimization literature [3], is to obtain a problem that may be written as a sum of individual terms by taking the dual of the inner supremum, yielding the convex problem

$$\inf_{x \in \mathcal{X}} \sup_{p \in \mathcal{P}_{\rho,n}} p^\top \ell(x) = \inf_{x \in \mathcal{X}, \lambda \geq 0, \eta \in \mathbb{R}} \frac{1}{n} \sum_{i=1}^n \lambda f^*\left(\frac{\ell_i(x) - \eta}{\lambda}\right) + \frac{\rho}{n}\lambda + \eta. \tag{5}$$

Here $f^*(s) = \sup_{t \geq 0}\{st - f(t)\}$ is the Fenchel conjugate of the convex function $f$. While the above dual reformulation is jointly convex in $(x, \lambda, \eta)$, canonical stochastic gradient descent (SGD) procedures [23] generally fail because the variance of the objective (and its subgradients) explodes as $\lambda \to 0$. (This is not just a theoretical issue: in extensive simulations that we omit because they are a bit boring, SGD and other heuristic approaches that impose shrinking bounds of the form $\lambda_t \geq c_t > 0$ at each iteration $t$ all fail to optimize the objective (5).)

Instead, we view the robust ERM problem (1) as a game between the $x$ (minimizing) player and $p$ (maximizing) player. Each player performs a variant of mirror descent (ascent), and we show how such an approach yields strong convergence guarantees, as well as good empirical performance. In particular, we show (for many suitable divergences $f$) that if $\ell_i$ is $L$-Lipschitz and $\mathcal{X}$ has radius bounded by $R$, then our procedure requires at most $O(\frac{R^2 L^2 + \rho}{\epsilon^2})$ iterations to achieve an $\epsilon$-accurate solution to problem (1), which is comparable to the number of iterations required by SGD [23]. Our solution strategy builds off of similar algorithms due to Nemirovski et al. [23, Sec. 3] and Ben-Tal et al. [4], and more directly procedures developed by Clarkson et al. [10] for solving two-player convex games. Most directly relevant to our approach is that of Shalev-Shwartz and Wexler [26], which solves problem (1) under the assumption that $\mathcal{P}_{\rho,n} = \{p \in \mathbb{R}^n_+ : p^T \mathbb{1} = 1\}$ and that there is some $x$ with perfect loss performance, that is, $\sum_{i=1}^n \ell_i(x) = 0$. We generalize these approaches to more challenging $f$-divergence-constrained problems, and, for the $\chi^2$ divergence with $f(t) = \frac{1}{2}(t-1)^2$,

develop efficient data structures that give a total run-time for solving problem (1) to $\epsilon$-accuracy scaling as $O((\mathsf{Cost}(\mathrm{grad}) + \log n)\frac{R^2 L^2 + \rho}{\epsilon^2})$. Here $\mathsf{Cost}(\mathrm{grad})$ is the cost to compute the gradient of a single term $\nabla \ell_i(x)$ and perform a mirror descent step with $x$. Using SGD to solve the empirical minimization problem to $\epsilon$-accuracy has run-time $O(\mathsf{Cost}(\mathrm{grad})\frac{R^2 L^2}{\epsilon^2})$, so we see that we can achieve the guarantees (3)–(4) offered by the robust formulation (1) at little additional computational cost.

The remainder of the paper is organized as follows. We present our abstract algorithm in Section 2 and give guarantees on its performance in Section 3. In Section 4, we give efficient computational schemes for the case that $f(t) = \frac{1}{2}(t-1)^2$, presenting experiments in Section 5.

## 2 A bandit mirror descent algorithm for the minimax problem

Under the conditions that $\ell$ is convex and $\mathcal{X}$ is compact, standard results [7] show that there exists a saddle point $(x^\star, p^\star) \in \mathcal{X} \times \mathcal{P}_{\rho,n}$ for the robust problem (1) satisfying

$$\sup\left\{p^\top \ell(x^\star) \mid p \in \mathcal{P}_{\rho,n}\right\} \le p^{\star\top} \ell(x^\star) \le \inf\left\{p^{\star\top} \ell(x) \mid x \in \mathcal{X}\right\}.$$

We now describe a procedure for finding this saddle point by alternating a linear bandit-convex optimization procedure [8] for $p$ and a stochastic mirror descent procedure for $x$. Our approach builds off of Nemirovski et al.'s [23] development of mirror descent for two-player stochastic games.

To describe our algorithm, we require a few standard tools. Let $\|\cdot\|_{\mathsf{x}}$ denote a norm on the space $\mathcal{X}$ with dual norm $\|y\|_{\mathsf{x},*} = \sup\{\langle x, y\rangle : \|x\| \le 1\}$, and let $\psi_{\mathsf{x}}$ be a differentiable strongly convex function on $\mathcal{X}$, meaning $\psi_{\mathsf{x}}(x + \Delta) \ge \psi_{\mathsf{x}}(x) + \nabla\psi_{\mathsf{x}}(x)^\top \Delta + \frac{1}{2}\|\Delta\|_{\mathsf{x}}^2$ for all $\Delta$. Let $\psi_{\mathsf{p}}$ a differentiable strictly convex function on $\mathcal{P}_{\rho,n}$. For a differentiable convex function $h$, we define the Bregman divergence $B_h(x, y) = h(x) - h(y) - \langle \nabla h(y), x - y\rangle 0$. The Fenchel conjugate $\psi_{\mathsf{p}}^*$ of $\psi_{\mathsf{p}}$ is

$$\psi_{\mathsf{p}}^*(s) := \sup_p\{\langle s, p\rangle - \psi_{\mathsf{p}}(p)\} \ \text{ and } \ \nabla\psi_{\mathsf{p}}^*(s) = \operatorname*{argmax}_p\{\langle s, p\rangle - \psi_{\mathsf{p}}(p)\}.$$

($\psi_{\mathsf{p}}^*$ is differentiable because $\psi_{\mathsf{p}}$ is strongly convex [20, Chapter X].) We let $g_i(x) \in \partial\ell_i(x)$ be a particular subgradient selection.

With this notation in place, we now give our algorithm, which alternates between gradient ascent steps on $p$ and subgradient descent steps on $x$. Roughly, we would like to alternate gradient ascent steps for $p$, $p_{t+1} \leftarrow p_t + \alpha_{\mathsf{p}}\ell(x_t)$, and descent steps $x_{t+1} \leftarrow x_t - \alpha_{\mathsf{x}}g_i(x_t)$ for $x$, where $i$ is a random index drawn according to $p_t$. This procedure is inefficient—requiring time of order $n\mathsf{Cost}(\mathrm{grad})$ in each iteration—so that we use stochastic estimates of the loss vector $\ell(x_t)$ developed in the linear bandit literature [8] and variants of mirror descent to implement our algorithm.

---

**Algorithm 1** Two-player Bandit Mirror Descent

---

1: Input: Stepsize $\alpha_{\mathsf{x}}, \alpha_{\mathsf{p}} > 0$, initialize: $x_1 \in \mathcal{X}$, $p_1 = \mathbb{1}/n$
2: **for** $t = 1, 2, \ldots, T$ **do**
3:     Sample $I_t \sim p_t$, that is, set $I_t = i$ with probability $p_{t,i}$
4:     Compute estimated loss for $i \in [n]$: $\widehat{\ell}_{t,i}(x) = \frac{\ell_i(x)}{p_{i,t}}\mathbb{1}\{I_t = i\}$
5:     Update $p$: $w_{t+1} \leftarrow \nabla\psi_{\mathsf{p}}^*(\nabla\psi_{\mathsf{p}}(p_t) + \alpha_{\mathsf{p}}\widehat{\ell}_t(x_t))$, $p_{t+1} \leftarrow \operatorname*{argmin}_{p \in \mathcal{P}_{\rho,n}} B_{\psi_{\mathsf{p}}}(p, w_{t+1})$
6:     Update $x$: $y_{t+1} \leftarrow \nabla\psi_{\mathsf{x}}^*(\psi_{\mathsf{x}}(x_t) - \alpha_x g_{I_t}(x_t))$,   $x_{t+1} \leftarrow \operatorname*{argmin}_{x \in \mathcal{X}} B_{\psi_{\mathsf{x}}}(x, y_{t+1})$
7: **end for**

---

We specialize this general algorithm for specific choices of the divergence $f$ and the functions $\psi_{\mathsf{x}}$ and $\psi_{\mathsf{p}}$ presently, first briefly discussing the algorithm. Note that in Step 5, the updates for $p$ depend only on a single index $I_t \in \{1, \ldots, n\}$ (the vector $\widehat{\ell}(x_t)$ is 1-sparse), which, as long as the updates for $p$ are efficiently computable, can yield substantial performance benefits.

## 3 Regret bounds

With our algorithm described, we now describe its convergence properties, specializing later to specific families of $f$-divergences. We begin with the following result on *pseudo*-regret, which (with minor modifications) is known [23, 10, 26]. We provide a proof for completeness in Appendix A.1.

**Lemma 1.** *Let the sequences $x_t$ and $p_t$ be generated by Algorithm 1. Define $\widehat{x}_T := \frac{1}{T}\sum_{t=1}^{T} x_t$ and $\widehat{p}_T := \frac{1}{T}\sum_{t=1}^{T} p_t$. Then for the saddle point $(x^\star, p^\star)$ we have*

$$T\mathbb{E}[p^{\star\top}\ell(\widehat{x}_T) - \widehat{p}_T^\top \ell(x^\star)] \leq \underbrace{\frac{1}{\alpha_{\mathsf{x}}} B_{\psi_{\mathsf{x}}}(x^\star, x_1) + \frac{\alpha_{\mathsf{x}}}{2}\sum_{t=1}^{T}\mathbb{E}[\|g_{I_t}(x_t)\|_{\mathsf{x},*}^2]}_{\mathcal{T}_1:\ \textit{ERM regret}} + \underbrace{\sum_{t=1}^{T}\mathbb{E}[\widehat{\ell}_t(x_t)^\top(p^\star - p_t)]}_{\mathcal{T}_2:\ \textit{robust regret}}$$

*where the expectation is taken over the random draws $I_t \sim p_t$. Moreover, $\mathbb{E}[\widehat{\ell}_t(x_t)^\top(p - p_t)] = \mathbb{E}[\ell(x_t)^\top(p - p_t)]$ for any vector $p$.*

In the lemma, $\mathcal{T}_1$ is the standard regret when applying mirror descent to the ERM problem. In particular, if $B_{\psi_{\mathsf{x}}}(x^\star, x_1) \leq R^2$ and $\ell_i(x)$ is $L$-Lipschitz, then choosing $\alpha_{\mathsf{x}} = \frac{R}{L}\sqrt{2/T}$ yields $\mathcal{T}_1 \leq RL\sqrt{T}$. Because it is (relatively) easy to bound the term $\mathcal{T}_1$, the remainder of our arguments focus on bounding the the second term $\mathcal{T}_2$, which is the regret that comes as a consequence of the random sampling for the loss vector $\widehat{\ell}_t$. This regret depends strongly on the distance-generating function $\psi_{\mathsf{p}}$. To the end of bounding $\mathcal{T}_2$, we use the following bound for the pseudo-regret of $p$, which is standard [9, Chapter 11], [8, Thm 5.3]. For completeness we outline the proof in Appendix A.2.

**Lemma 2.** *For any $p \in \mathcal{P}_{\rho,n}$, Algorithm 1 satisfies*

$$\sum_{t=1}^{T}\widehat{\ell}_t(x_t)^\top(p - p_t) \leq \frac{B_{\psi_{\mathsf{p}}}(p, p_1)}{\alpha_{\mathsf{p}}} + \frac{1}{\alpha_{\mathsf{p}}}\sum_{t=1}^{T} B_{\psi_{\mathsf{p}}^*}\left(\nabla\psi_{\mathsf{p}}(p_t) + \alpha_{\mathsf{p}}\widehat{\ell}_t(x_t), \nabla\psi_{\mathsf{p}}(p_t)\right). \quad (6)$$

Lemma 2 shows that controlling the Bregman divergences $B_{\psi_{\mathsf{p}}}$ and $B_{\psi_{\mathsf{p}}^*}$ is sufficient to bound $\mathcal{T}_2$ in the basic regret bound of Lemma 1.

Now, we narrow our focus slightly to a specialized—but broad—family of divergences for which we can give more explicit results. For $k \in \mathbb{R}$, the Cressie-Read divergence [12] of order $k$ is

$$f_k(t) = \frac{t^k - kt + k - 1}{k(k-1)}, \quad (7)$$

where $f_k(t) = \infty$ for $t < 0$, and for $k \in \{0, 1\}$ we define $f_k$ by its limits as $k \to 0$ or $1$ (we have $f_1(t) = t\log t - t + 1$ and $f_0(t) = -\log t + t - 1$). Inspecting expression (6), we might hope that careful choices of $\psi_{\mathsf{p}}$ could yield regret bounds that grow slowly with $T$ and have small dependence on the sample size $n$. Indeed, this is the case, as we show in the sequel: for each divergence $f_k$, we may carefully choose $\psi_{\mathsf{p}}$ to achieve small regret. To prove our bounds, however, it is crucial that the importance sampling estimator $\widehat{\ell}_t$ has small variance, which in turn necessitates that $p_{t,i}$ is not too small. Generally, this means that in the update (Alg. 1, Line 5) to construct $p_{t+1}$, we choose $\psi(p)$ to grow quickly as $p_i \to 0$ (e.g. $|\frac{\partial}{\partial p_i}\psi_{\mathsf{p}}(p)| \to \infty$), but there is a tradeoff in that this may cause large Bregman divergence terms (6). In the coming sections, we explore this tradeoff for various $k$, providing regret bounds for each of the Cressie-Read divergences (7).

To control the $B_{\psi_{\mathsf{p}}^*}$ terms in the bound (6), we use the curvature of $\psi_{\mathsf{p}}$ (dually, smoothness of $\psi_{\mathsf{p}}^*$) to show that $B_{\psi_{\mathsf{p}}^*}(u, v) \approx \sum(u_i - v_i)^2$. For this approximation to hold, we shift our loss functions based on the $f$-divergence. When $k \geq 2$, we assume that $\ell(x) \in [0,1]^n$. If $k < 2$, we instead apply Algorithm 1 with shifted losses $\ell'(x) = \ell(x) - \mathbb{1}$, so that $\ell'(x) \in [-1,0]^n$. We call the method with $\ell'$ Algorithm 1', noting that $\widehat{\ell}_{t,i}(x_t) = \frac{\ell_i(x_t)-1}{p_{t,i}}\mathbb{1}\{I_t = i\}$ in this case.

## 3.1 Power divergences when $k \notin \{0, 1\}$

For our first results, we prove a generic regret bound for Algorithm 1 when $k \notin \{0, 1\}$ by taking the distance-generating function $\psi_{\mathsf{p}}(p) = \frac{1}{k(k-1)}\sum_{i=1}^{n} p_i^k$, which is differentiable and strictly convex on $\mathbb{R}_+^n$. Before proceeding further, we first note that for $p \in \mathcal{P}_{\rho,n}$ and $p_1 = \frac{1}{n}\mathbb{1}$, we have

$$B_{\psi_{\mathsf{p}}}(p, p_1) = \psi_{\mathsf{p}}(p) - \psi_{\mathsf{p}}(p_1) - \nabla\psi_{\mathsf{p}}(p_1)^\top(p - p_1)$$

$$= \frac{n^{-k}}{k(k-1)}\sum_{i=1}^{n}\left\{(np_i)^k - knp_i + k - 1\right\} = n^{-k}D_f(p\|\mathbb{1}/n) \leq n^{-k}\rho \quad (8)$$

bounding the first term in expression (6). From Lemma 2, it remains to bound the Bregman divergence terms $B_{\psi_{\mathsf{p}}^*}$. Using smoothness of $\psi_{\mathsf{p}}^*$ in the positive orthant, we obtain the following bound.

**Theorem 1.** *Assume that $\ell(x) \in [0,1]^n$. For any real-valued $k \geq 2$ and any $p \in \mathcal{P}_{\rho,n}$, Algorithm 1 satisfies*

$$\sum_{t=1}^{T} \mathbb{E}[\ell(x_t)^\top (p - p_t)] = \sum_{t=1}^{T} \mathbb{E}[\widehat{\ell}_t(x_t)^\top (p - p_t)] \leq \frac{n^{-k}\rho}{\alpha_{\mathsf{p}}} + \frac{\alpha_{\mathsf{p}}}{2} \sum_{t=1}^{T} \mathbb{E}\left[ \sum_{i:p_{t,i}>0} p_{t,i}^{1-k} \right]. \qquad (9)$$

*For $k \leq 2$ with $k \notin \{0,1\}$, an identical bound holds for Algorithm 1' with $\ell'(x) = \ell(x) - \mathbb{1}$.*

See Appendix A.3 for the proof. We now use Theorem 1 to obtain concrete convergence guarantees for Cressie-Read divergences with parameter $k < 1$, giving sublinear (in $T$) regret bounds independent of $n$. In the corollary, whose proof we provide in Appendix A.4, we let $C_{k,\rho} = \frac{(1-k)(1-k\rho)}{-k}$, which is positive for $k < 0$.

**Corollary 1.** *For $k \in (-\infty, 0)$ and $\alpha_{\mathsf{p}} = C_{k,\rho}^{\frac{k-1}{2}} n^{-k} \sqrt{2\rho/T}$ Algorithm 1' with $\ell'(x) = \ell(x) - \mathbb{1} \in [-1,0]^n$ acheives the regret bound*

$$\sum_{t=1}^{T} \mathbb{E}[\ell(x_t)^\top (p - p_t)] = \sum_{t=1}^{T} \mathbb{E}[\widehat{\ell}_t(x_t)^\top (p - p_t)] \leq \sqrt{2C_{k,\rho}^{1-k}\rho T}.$$

*For $k \in (0,1)$ and $\alpha_{\mathsf{p}} = n^{-k}\sqrt{2\rho/T}$, Algorithm 1' with $\ell'(x) = \ell(x) - \mathbb{1} \in [-1,0]^n$ acheives the regret bound*

$$\sum_{t=1}^{T} \mathbb{E}[\ell(x_t)^\top (p - p_t)] = \sum_{t=1}^{T} \mathbb{E}[\widehat{\ell}_t(x_t)^\top (p - p_t)] \leq \sqrt{2\rho T}.$$

It is worth noting that despite the robustification, the above regret is independent of $n$. In the special case that $k \in (0,1)$, Theorem 1 is the regret bound for the implicitly normalized forecaster of Audibert and Bubeck [1] (cf. [8, Ch 5.4]).

## 3.2 Regret bounds using the KL divergences ($k = 1$ and $k = 0$)

The choice $f_1(t) = t \log t - t + 1$ yields $D_f(P\|Q) = D_{\mathrm{kl}}(P\|Q)$, and in this case, we take $\psi_{\mathsf{p}}(p) = \sum_{i=1}^{n} p_i \log p_i$, which means that Algorithm 1 performs entropic gradient ascent. To control the divergence $B_{\psi_{\mathsf{p}}^*}$, we use the rescaled losses $\ell'(x) = \ell(x) - \mathbb{1}$ (as we have $k < 2$). Then we have the following bound, whose proof we provide in Appendix A.5.

**Theorem 2.** *Algorithm 1' with loss $\ell'(x) = \ell(x) - \mathbb{1}$ yields*

$$\sum_{t=1}^{T} \mathbb{E}[\ell(x_t)^\top (p - p_t)] = \sum_{t=1}^{T} \mathbb{E}[\widehat{\ell}_t(x_t)^\top (p - p_t)] \leq \frac{\rho}{n\alpha_{\mathsf{p}}} + \frac{\alpha_{\mathsf{p}}}{2} nT. \qquad (10)$$

*In particular, when $\alpha_{\mathsf{p}} = \frac{1}{n}\sqrt{\frac{2\rho}{T}}$, we have $\sum_{t=1}^{T} \mathbb{E}[\ell(x_t)^\top (p - p_t)] \leq \sqrt{2\rho T}$.*

Using $k = 0$, so that $f_0(t) = -\log t + t - 1$, we obtain $D_f(P\|Q) = D_{\mathrm{kl}}(Q\|P)$, which results in a robustification technique identical to Owen's original empirical likelihood [24]. We again use the rescaled losses $\ell'(x) = \ell(x) - \mathbb{1}$, but in this scenario we use the proximal function $\psi_{\mathsf{p}}(p) = -\sum_{i=1}^{n} \log p_i$ in Algorithm 1'. Then we have the following regret bound (see Appendix A.6).

**Theorem 3.** *Algorithm 1' with loss $\ell'(x) = \ell(x) - \mathbb{1}$ yields*

$$\sum_{t=1}^{T} \mathbb{E}[\ell(x_t)^\top (p - p_t)] = \sum_{t=1}^{T} \mathbb{E}[\widehat{\ell}_t(x_t)^\top (p - p_t)] \leq \frac{\rho}{\alpha_{\mathsf{p}}} + \frac{\alpha_{\mathsf{p}}}{2} T.$$

*In particular, when $\alpha_{\mathsf{p}} = \sqrt{\frac{2\rho}{T}}$, we have $\sum_{t=1}^{T} \mathbb{E}[\ell(x_t)^\top (p - p_t)] \leq \sqrt{2\rho T}$.*

In both of these cases, the expected pseudo-regret of our robust gradient procedure is independent of $n$ and grows as $\sqrt{T}$, which is essentially identical to that achieved by pure online gradient methods.

### 3.3 Power divergences ($k > 1$)

Corollary 1 provides convergence guarantees for power divergences $f_k$ with $k < 1$, but says nothing about the case that $k > 1$; the choice $\psi_{\mathsf{p}}(p) = \frac{1}{k(k-1)} \sum_{i=1}^{n} p_i^k$ allows the individual probabilities $p_{t,i}$ to be too small, which can cause excess variance of $\widehat{\ell}$. To remedy this, we regularize the robust problem (1) by re-defining our robust empirical distributions set, taking

$$\mathcal{P}_{\rho,n,\delta} := \Big\{ p \in \mathbb{R}_+^n \mid p \geq \frac{\delta}{n}, \sum_{i=1}^{n} f(np_i) \leq \rho \Big\},$$

where we no longer constrain the weights $p$ to satisfy $\mathbb{1}^\top p = 1$. Nonetheless, it is still possible to show that the guarantees (2) and (3) hold with $\mathcal{P}_{\rho,n,\delta}$ replacing $\mathcal{P}_{\rho,n}$. Indeed, we may give bounds for the pseudo-regret of the regularized problem with $\mathcal{P}_{\rho,n,\delta}$, where we apply Algorithm 1 with a slightly modified sampling strategy, drawing indices $i$ according to the normalized distribution $p_t / \sum_{i=1}^{n} p_{t,i}$ and appropriately normalizing the loss estimate via

$$\widehat{\ell}_{t,i}(x_t) = \Big( \sum_{i=1}^{n} p_{t,i} \Big) \frac{\ell_i(x_t)}{p_{t,i}} \mathbf{1}\{I_t = i\}.$$

This vector is still unbiased for $\ell(x_t)$. Define the constant $C_k := \max\{t : f_k(t) \leq t\} \vee \frac{\rho}{n} < \infty$ (so $C_2 = 2 + \sqrt{3}$). With our choice $\psi_{\mathsf{p}}(p) = \frac{1}{k(k-1)} \sum_{i=1}^{n} p_i^k$ and for $\delta > 0$, we obtain the following result, whose proof we provide in Appendix A.7.

**Theorem 4.** *For $k \in [2, \infty)$, any $p \in \mathcal{P}_{\rho,n,\delta}$, Algorithm 1 with $\alpha_{\mathsf{p}} = n^{-k}\sqrt{\rho \delta^{k-1}/(4C_k^3 T)}$ yields*

$$\sum_{t=1}^{T} \mathbb{E}[\ell(x_t)^\top (p - p_t)] = \sum_{t=1}^{T} \mathbb{E}[\widehat{\ell}_t(x_t)^\top (p - p_t)] \leq 2C_k \sqrt{\rho C_k \delta^{1-k} T}$$

*For $k \in (1, 2)$, assume that $\ell(x) \in [-1, 0]^n$. Then, Algorithm 1 gives identical bounds.*

## 4 Efficient updates when $k = 2$

The previous section shows that Algorithm 1 with careful choice of $\psi_{\mathsf{p}}$ yields sublinear regret bounds. The projection step $p_{t+1} = \operatorname{argmin}_{p \in \mathcal{P}_{\rho,n,\delta}} B_{\psi_{\mathsf{p}}}(p, w_{t+1})$, however, can still take time linear in $n$ despite the sparsity of $\widehat{\ell}(x_t)$ (see Appendix B for concrete updates for each of our cases). In this section, we show how to compute the bandit mirror descent update in Alg. 1, line 5, in time $O(\log n)$ time for $f_2(t) = \frac{1}{2}(t-1)^2$ and $\psi_{\mathsf{p}}(p) = \frac{1}{2} \sum_{i=1}^{n} p_i^2$. Building off of Duchi et al. [14], we use carefully designed balanced binary search trees (BSTs) to this end.

The Lagrangian for the update $p_{t+1} = \operatorname{argmin}_{p \in \mathcal{P}_{\rho,n,\delta}} B_{\psi_{\mathsf{p}}}(p, w_{t+1})$ (suppressing $t$) is

$$\mathcal{L}(p, \lambda, \theta) = B_{\psi_{\mathsf{p}}}(p, w) - \frac{\lambda}{n^2}\Big( \rho - \sum_{i=1}^{n} f_2(np_i) \Big) - \theta^\top \Big( p - \frac{\delta}{n}\mathbb{1} \Big)$$

where $\lambda \geq 0, \theta \in \mathbb{R}_+^n$. The KKT conditions imply $(1+\lambda)p = w + \frac{\lambda}{n}\mathbb{1} + \theta$, and strict complementarity yields

$$p(\lambda) = \Big( \frac{1}{1+\lambda}w + \frac{\lambda}{1+\lambda}\frac{1}{n} - \frac{\delta}{n}\mathbb{1} \Big)_+ + \frac{\delta}{n}\mathbb{1}, \tag{11}$$

where $p(\lambda) = \operatorname{argmin}_{p \in \mathcal{P}_{\rho,n,\delta}} \inf_{\theta \in \mathbb{R}_+^n} \mathcal{L}(p, \lambda, \theta)$. Substituting this into the Lagrangian, we obtain the concave dual objective

$$g(\lambda) := \sup_{\theta} \inf_{p \in \mathcal{P}_{\rho,n,\delta}} \mathcal{L}(p, \lambda, \theta) = B_{\psi_{\mathsf{p}}}(p(\lambda), w) - \lambda \Big( \rho - \sum_{i=1}^{n} f_k(np_i(\lambda)) \Big).$$

We can run a bisection search on the nondecreasing function $g'(\lambda)$ to find $\lambda$ such that $g'(\lambda) = 0$. After algebraic manipulations, we have that

$$\frac{\partial}{\partial \lambda}g(\lambda) = g_1(\lambda) \sum_{i \in I(\lambda)} w_i^2 + g_2(\lambda) \sum_{i \in I(\lambda)} w_i + g_3(\lambda)|I(\lambda)| + \frac{(1-\delta)^2}{2n} - \frac{\rho}{n^2},$$

where $I(\lambda) := \{1 \le i \le n : w_i \ge \frac{\delta}{n} + (\frac{\delta}{n} - 1)\lambda\}$ and (see expression (18) in Appendix B.4)

$$g_1(\lambda) = \frac{1}{(1+\lambda)^2}, \quad g_2(\lambda) = \frac{-2}{n(1+\lambda)^2}, \quad g_3(\lambda) = \frac{1}{n^2(1+\lambda)^2} - \frac{(1-\delta)^2}{2n}.$$

To see that we can solve for $\lambda^*$ that acheives $|g'(\lambda^*)| \le \epsilon$ in $O(\log n + \log \frac{1}{\epsilon})$ time, it suffices to evaluate $\sum_{i \in I(\lambda)} w_i^q$ for $q = 0, 1, 2$ in time $O(\log n)$. To this end, we store the $w$'s in a balanced search tree (*e.g.*, red-black tree) keyed on the weights up to a multiplicative and an additive constant. A key ingredient in our implementation is that the BST stores in each node the sum of the appropriate powers of values in the left and right subtree [14]. See Appendix C for detailed pseudocode for all operations required in Algorithm 1: each subroutine (sampling $I_t \sim p_t$, updating $w$, computing $\lambda^*$, and updating $p(\lambda^*)$) require time $O(\log n)$ using standard BST operations.

## 5    Experiments

In this section, we present experimental results demonstrating the efficiency of our algorithm. We first compare our method with existing algorithms for solving the robust problem (1) on a synthetic dataset, then investigating the robust formulation on real datasets to show how the calibrated confidence guarantees behave in practice, especially in comparison to the ERM. We experiment on natural high dimensional datasets as well as those with many training examples.

Our implementation uses the efficient updates outlined in Section 4. Throughout our experiments, we use the best tuned step sizes for all methods. For the first two experiments, we set $\rho = \chi_{1,.9}^2$ so that the resulting robust objective (1) will be a calibrated $95\%$ upper confidence bound on the optimal population risk. For our last experiment, the asymptotic regime (3) fails to hold due to the high dimensional nature of the problem, so we choose $\rho = 50$ (somewhat arbitrarily, but other $\rho$ give similar behavior). We take $\mathcal{X} = \{x \in \mathbb{R}^d : \|x\|_2 \le R\}$ for our experiments.

For the experiment with synthetic data, we compare our algorithm against two benchmark methods for solving the robust problem (1). The first is the interior point method for the dual reformulation (5) using the Gurobi solver [17]. The second is using gradient descent, viewing the robust formulation (1) as a minimization problem with the objective $x \mapsto \sup_{p \in \mathcal{P}_{\rho,n,\delta}} p^\top \ell(x)$. To efficiently compute the gradient, we bisect over the dual form (5) with respect to $\lambda \ge 0, \eta$. We use the best step sizes for both our proposed bandit-based algorithm and gradient descent.

To generate the data, we choose a true classifier $x^* \in \mathbb{R}^d$ and sample the feature vectors $a_i \overset{\text{iid}}{\sim} \mathsf{N}(0, I)$ for $i \in [n]$. We set the labels to be $b_i = \text{sign}(a_i^\top x^*)$ and flip them with probability $10\%$. We use the hinge loss $\ell_i(x) = (1 - b_i a_i^\top x)_+$ with $n = 2000$, $d = 500$ and $R = 10$ in our experiment. In Figure 1a, we plot the log optimality ratio (log of current objective value over optimal value) with respect to the runtime for the three algorithms. While the interior point method (IPM) obtains accurate solutions, it scales relatively poorly in $n$ and $d$ (the initial flat region in the plot is due to pre-computations for factorizing within the solver). Gradient descent performs quite well in this moderate sized example although each iteration takes time $\Omega(n)$.

We also perform experiments on two datasets with larger $n$: the Adult dataset [22] and the Reuters RCV1 Corpus [21]. The Adult dataset has $n = 32,561$ training and $16,281$ test examples with 123-dimensional features. We use binary logistic loss $\ell_i(x) = \log(1 + \exp(-b_i a_i^\top x))$ to classify whether the income level is greater than \$5K. For the Reuters RCV1 Corpus, our task is to classify whether a document belongs to the Corporate category. With $d = 47,236$ features, we randomly split the $804,410$ examples into $723,969$ training ($90\%$ of data) and $80,441$ ($10\%$ of data) test examples. We use the hinge loss and solve the binary classification problem for the document type. To test the efficiency of our method in large scale settings, we plot the log ratio $\log \frac{R_n(x)}{R_n(x^*)}$, where $R_n(x) = \sup_{p \in \mathcal{P}_{\rho,n,\delta}} p^\top \ell(x)$, versus CPU time for our algorithm and gradient descent in Figure 1b. As is somewhat typical of stochastic gradient-based methods, our bandit-based optimization algorithm quickly obtains a solution with small optimality gap (about $2\%$ relative error), while the gradient descent method eventually achieves better loss.

In Figures 2a–2d, we plot the loss value and the classification error compared with applying pure stochastic gradient descent to the standard empirical loss, plotting the confidence bound for the robust

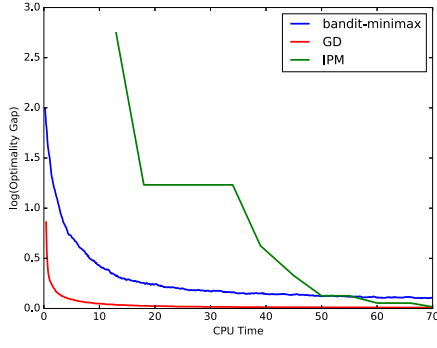
(a) Synthetic Data ($n = 2000, d = 500$)

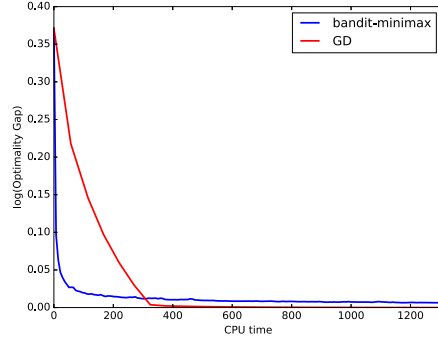
(b) Reuters Corpus ($n = 7.2 \cdot 10^5, d \approx 5 \cdot 10^4$)

Figure 1: Comparison of Solvers

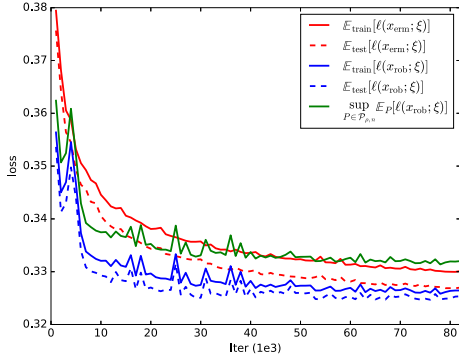
(a) Adult: Logistic Loss

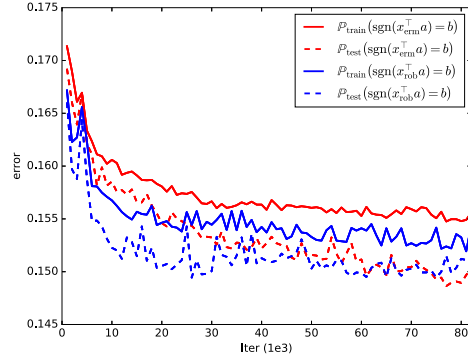
(b) Adult: Classification Error

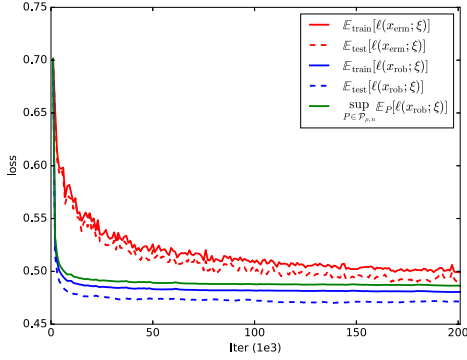
(c) Reuters: Hinge Loss

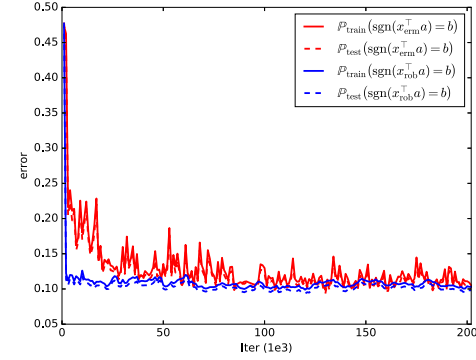
(d) Reuters: Classification Error

Figure 2: Comparison with ERM

method as well. As the theory suggests [15, 13], the robust objective provides upper confidence bounds on the true risk (approximated by the average loss on the test sample).

## Acknowledgments

JCD and HN were partially supported by the SAIL-Toyota Center for AI Research and the National Science Foundation award NSF-CAREER-1553086. HN was also partially supported Samsung Fellowship.

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
