[Supplementary Material · supplement.pdf]

# A Proofs of Regret Bounds

## A.1 Proof of Lemma 1

From convexity of the loss function $\ell(\cdot)$, we have

$$p^\top \ell(\widehat{x}_T) - \widehat{p}_T^\top \ell(x) \leq \frac{1}{T} \sum_{t=1}^{T} p^\top \ell(x_t) - p_t^\top \ell(x) \leq \frac{1}{T} \sum_{t=1}^{T} \left\{ \ell(x_t)^\top (p - p_t) + p_t^\top g(x_t)(x - x_t) \right\} \tag{12}$$

where we have used $g(x_t) \in R^{n \times d}$ to denote the $n$-by-$d$ matrix whose rows are $g_i(x_t)^\top$. Note that $\mathbb{E}[\widehat{\ell}_t(x_t)^\top (p - p_t)|I_1^{t-1}, x_1^t] = \ell(x_t)^\top (p - p_t)$, $\mathbb{E}[g_{I_t}(x_t)^\top (x - x_t)|I_1^{t-1}, x_1^t] = p_t^\top g(x_t)(x - x_t)$ since $I_t \sim p_t$ and $p_t$ is $\sigma(I_1^{t-1}, x_1^t)$-measurable. Further, from the standard mirror descent result (*e.g.*, [23, Section 2.3]), we have

$$\sum_{t=1}^{T} \mathbb{E}[g_{I_t}(x_t)^\top (x - x_t)] \leq \frac{1}{\alpha_x} B_{\psi_x}(x^\star, x_1) + \frac{\alpha_x}{2} \sum_{t=1}^{T} \mathbb{E} \|g_{I_t}(x_t)\|_{x,*}^2.$$

Taking expectation in (12) and applying these facts, desired result follows.

## A.2 Proof of Lemma 2

From Algorithm 1, we have

$$\alpha_{\mathsf{p}} \widehat{\ell}_t(x_t)^\top (p - p_t) = (\nabla \psi_{\mathsf{p}}(w_{t+1}) - \nabla \psi_{\mathsf{p}}(x_t))^\top (p - p_t)$$
$$= B_{\psi_{\mathsf{p}}}(p, p_t) + B_{\psi_{\mathsf{p}}}(p_t, w_{t+1}) - B_{\psi_{\mathsf{p}}}(p, w_{t+1}). \tag{13}$$

For any $p \in \mathcal{P}_{\rho,n}$, we have for all $p \in \mathcal{P}_{\rho,n}$,

$$B_{\psi_{\mathsf{p}}}(p, w_{t+1}) \geq B_{\psi_{\mathsf{p}}}(p, p_{t+1}) + B_{\psi_{\mathsf{p}}}(p_{t+1}, w_{t+1}) \equiv (\nabla \psi_{\mathsf{p}}(p) - \nabla \psi_{\mathsf{p}}(w_{t+1}))^\top (p - p_{t+1}) \geq 0.$$

The latter inequality is just the optimality condition for $p_{t+1} = \operatorname{argmin}_{p \in \mathcal{P}_{\rho,n}} B_{\psi_{\mathsf{p}}}(p, w_{t+1})$. Applying the first equality in (13) and summing for $t = 1, \ldots, T$, we obtain

$$\alpha_{\mathsf{p}} \sum_{t=1}^{T} \widehat{\ell}_t(x_t)^\top (p - p_t) \leq B_{\psi_{\mathsf{p}}}(p, p_1) - B_{\psi_{\mathsf{p}}}(p, p_{T+1}) + \sum_{t=1}^{T} \left( B_{\psi_{\mathsf{p}}}(p_t, w_{t+1}) - B_{\psi_{\mathsf{p}}}(p_{t+1}, w_{t+1}) \right)$$

$$\leq B_{\psi_{\mathsf{p}}}(p, p_1) + \sum_{t=1}^{T} B_{\psi_{\mathsf{p}}}(p_t, w_{t+1})$$

$$= B_{\psi_{\mathsf{p}}}(p, p_1) + \sum_{t=1}^{T} B_{\psi_{\mathsf{p}}^*}(\nabla \psi_{\mathsf{p}}(w_{t+1}), \nabla \psi_{\mathsf{p}}(p_t)).$$

Now, noting that $\nabla \psi_{\mathsf{p}}(w_{t+1}) = \nabla \psi_{\mathsf{p}}(p_t) + \alpha_{\mathsf{p}} \widehat{\ell}_t(x_t)$, we obtain the result.

## A.3 Proof of Theorem 1

The conjugate of $\psi_{\mathsf{p}}$ is

$$\psi_{\mathsf{p}}^*(s) = \frac{1}{k} \left( (k-1)s \right)_+^{k_*} + \infty \cdot \mathbf{1} \left\{ s \leq 0, k < 1 \right\}.$$

From Taylor's theorem, we have

$$B_{\psi_{\mathsf{p}}^*}(u, v) = \frac{1}{k} \sum_{i=1}^{n} \left( ((k-1)u_i)_+^{k_*} - ((k-1)v_i)_+^{k_*} \right) - \sum_{i=1}^{n} ((k-1)v_i)_+^{k_*-1} (u_i - v_i)$$

$$= \sum_{i=1}^{n} \left( \int_{v_i}^{u_i} ((k-1)t)_+^{k_*-1} dt - ((k-1)v_i)_+^{k_*-1} (u_i - v_i) \right)$$

$$\leq \frac{1}{2} \sum_{i=1}^{n} \max_{t \in [v_i, u_i]} ((k-1)t)_+^{k_*-2} (u_i - v_i)^2$$

which gives the following useful lemma.

**Lemma 3** (Bubeck and Cesa-Bianchi [8], Lemma 5.9)**.**

$$B_{\psi_\mathsf{p}^*}(u, v) \leq \frac{1}{2} \sum_{i=1}^{n} \max_{t \in [v_i, u_i]} ((k-1)t)_+^{k_*-2} (u_i - v_i)^2.$$

For later use, we define the conjugate $k_* = \frac{k}{k-1}$ and note that

$$k_* = \frac{k}{k-1} = \begin{cases} < 1 & \text{if } k \in (-\infty, 0) \\ < 0 & \text{if } k \in (0, 1) \\ > 0 & \text{if } k \in (1, 2) \\ < 2 & \text{if } k \in (2, \infty) \end{cases}$$

Now, define $u := \nabla \psi_\mathsf{p}(p_t) + \alpha_\mathsf{p} \widehat{\ell}_t(x_t) = \frac{1}{k-1} p_t^{k-1} + \alpha_\mathsf{p} \widehat{\ell}_t(x_t)$ and $v := \nabla \psi_\mathsf{p}(p_t) = \frac{1}{k-1} p_t^{k-1}$, where $p^{k-1}$ indicates the vector with each of its entries raised to the power $k - 1$.

When $k \geq 2$, we have from Lemma 3 that

$$B_{\psi_\mathsf{p}^*}\left(\nabla \psi_\mathsf{p}(p_t) + \alpha_\mathsf{p} \widehat{\ell}_t(x_t), \nabla \psi_\mathsf{p}(p_t)\right) \leq \frac{p_{t, I_t}^{2-k}}{2} \left(\alpha_\mathsf{p} \frac{\ell_{I_t}(x_t)}{p_{t, I_t}}\right)^2 \leq \frac{\alpha_\mathsf{p}^2}{2} p_{t, I_t}^{-k} \tag{14}$$

where we have used that $\ell_i(x) \in [0, 1]$. Substituting this in the bound (6) and taking expectations, we obtain the result by noting that $I_t \sim p_t$.

For $k < 2$, note that since $p, p_t$ are probability vectors and $p_t$ is $\sigma(I_1^{t-1}, x_1^t)$-measurable, we have

$$\mathbb{E}[\widehat{\ell}_t(x_t)^\top (p - p_t) | I_1^{t-1}, x_1^t] = \ell'(x_t)^\top (p - p_t) = (\ell(x_t) - \mathbb{1})^\top (p - p_t) = \ell(x_t)^\top (p - p_t) \tag{15}$$

from which the first equality of the theorem follows. Following the proof of Lemma 2 verbatim, we have the usual regret bound

$$\sum_{t=1}^{T} \widehat{\ell}_t(x_t)^\top (p - p_t) \leq \frac{B_{\psi_\mathsf{p}}(p, p_1)}{\alpha_\mathsf{p}} + \frac{1}{\alpha_\mathsf{p}} \sum_{t=1}^{T} B_{\psi_\mathsf{p}^*}\left(\nabla \psi_\mathsf{p}(p_t) + \alpha_\mathsf{p} \widehat{\ell}_t(x_t), \nabla \psi_\mathsf{p}(p_t)\right) \tag{16}$$

where we now have $\widehat{\ell}_{t,i}(x_t) = \frac{\ell_i'(x_t)}{p_{t,i}} \mathbf{1}\{I_t = i\} = \frac{\ell_i(x_t)-1}{p_{t,i}} \mathbf{1}\{I_t = i\}$. Now, note that if $k \leq 2$ with $k \notin \{0, 1\}$, we have that $((k-1)s)_+^{k_*-2}$ is nondecreasing in $s$. Hence, we again obtain the bound (14) from Lemma 3.

## A.4 Proof of Corollary 1

When $k \in (-\infty, 0)$, the $f$-divergence constraint

$$\frac{1}{nk(k-1)} \sum_{i=1}^{n} \left\{(np_i)^k - k(np_i - 1) - 1\right\} \leq \frac{\rho}{n}$$

implies that $-knp_i \leq (np_i)^k - knp_i \leq (1-k)(1-k\rho)$ and hence $p_i \leq \frac{C_k}{n}$. Using this to bound the sum in (9), we get

$$\sum_{t=1}^{T} \mathbb{E}[\widehat{\ell}_t(x_t)^\top (p - p_t)] \leq \frac{n^{-k} \rho}{\alpha_\mathsf{p}} + \frac{\alpha_\mathsf{p}}{2} Tn^k C_k^{1-k}.$$

Minimizing with respect to $\alpha_\mathsf{p} > 0$ gives the first result. When $k \in (0, 1)$, we use Holder inequality with $p = \frac{1}{1-k} > 1$ and $q = \frac{1}{k} > 1$:

$$\sum_{i=1}^{n} p_{t,i}^{1-k} \leq \left(\sum_{i=1}^{n} (p_{t,i}^{1-k})^{\frac{1}{1-k}}\right)^{1-k} \left(\sum_{i=1}^{n} 1\right)^k = n^k.$$

Applying this bound in (9) and minimizing with respect to $\alpha_\mathsf{p}$, result follows.

## A.5 Proof of Theorem 2

Proceeding as in Section A.3 for $k \leq 2$, we obtain the regret bound (16). First, note that $B_{\psi_{\mathsf{p}}}(p, p_1) = \sum_{i=1}^{n} p_i \log n p_i \leq \frac{\rho}{n}$. We now bound $B_{\psi_{\mathsf{p}}^*}\left(\nabla \psi_{\mathsf{p}}(p_t) + \alpha_{\mathsf{p}} \widehat{\ell}_t(x_t), \nabla \psi_{\mathsf{p}}(p_t)\right)$. Using $\exp(-x) - 1 + x \leq \frac{x^2}{2}$ for $x \geq 0$, we have

$$B_{\psi_{\mathsf{p}}^*}\left(\log p_t + \mathbb{1} + \alpha_{\mathsf{p}} \widehat{\ell}_t(x_t), \log p_t + \mathbb{1}\right)$$

$$= \sum_{i \neq I_t} \exp(\log p_{t,i}) + \exp(\log p_{t,I_t} + \alpha_{\mathsf{p}} \widehat{\ell}_{t,I_t}(x_t)) - \sum_{i=1}^{n} \exp(\log p_{t,i}) - \exp(\log p_{t,I_t}) \alpha_{\mathsf{p}} \widehat{\ell}_{t,I_t}(x_t)$$

$$= p_{t,I_t} \left\{ \exp(\alpha_{\mathsf{p}} \widehat{\ell}_{t,I_t}(x_t)) - 1 - \alpha_{\mathsf{p}} \widehat{\ell}_{t,I_t}(x_t) \right\} \leq \frac{1}{2} p_{t,I_t} \widehat{\ell}_{t,I_t}(x_t)^2 = \frac{(\ell_{I_t}(x_t) - 1)^2}{2 p_{t,I_t}}$$

where we used $x = -\alpha_{\mathsf{p}} \widehat{\ell}_{t,I_t}(x_t) \geq 0$. Plugging the above observations into (6) and taking expectations, we obtain

$$\sum_{t=1}^{T} \mathbb{E}[\widehat{\ell}_t(x_t)^\top (p - p_t)] \leq \frac{\rho}{n \alpha_{\mathsf{p}}} + \frac{\alpha_{\mathsf{p}}}{2} \sum_{t=1}^{T} \mathbb{E}\left[\sum_{i=1}^{n} (\ell_i(x_t) - 1)^2\right].$$

Bounding $(\ell_i(x_t) - 1)^2 \leq 1$, the first claim follows. Optimizing the bound with respect to $\alpha_{\mathsf{p}} > 0$, we obtain the second result.

## A.6 Proof of Theorem 3

As in Section A.3, the first equality and the interim regret bound follows from (15), (16). Now, note that $B_{\psi_{\mathsf{p}}}(p, p_1) = -\sum_{i=1}^{n} (\log(n p_i) - n p_i + 1) \leq \rho$ to bound the first term. Next, we use $x - \log(1 + x) \leq \frac{x^2}{2}$ for $x \geq 0$ to get

$$B_{\psi_{\mathsf{p}}^*}\left(\nabla \psi_{\mathsf{p}}(p_t) + \alpha_{\mathsf{p}} \widehat{\ell}_t(x_t), \nabla \psi_{\mathsf{p}}(p_t)\right) = B_{\psi_{\mathsf{p}}^*}\left(-\frac{1}{p_t} + \alpha_{\mathsf{p}} \widehat{\ell}_t(x_t), -\frac{1}{p_t}\right)$$

$$= -\log\left(1 - \alpha_{\mathsf{p}} \ell'_{I_t}(x_t)\right) - \alpha_{\mathsf{p}} \ell'_{I_t}(x_t) \leq \frac{\alpha_{\mathsf{p}}^2 \ell'_{I_t}(x_t)^2}{2}$$

where we have used $x = -\alpha_{\mathsf{p}} \ell'_{I_t}(x_t) \geq 0$ and $\ell'_{I_t}(x_t) \in [-1, 0]$. Plugging these into the bound (6) and taking expectations, we have

$$\sum_{t=1}^{T} \mathbb{E}[\widehat{\ell}_t(x_t)^\top (p - p_t)] \leq \frac{\rho}{\alpha_{\mathsf{p}}} + \frac{\alpha_{\mathsf{p}}}{2} \sum_{t=1}^{T} \sum_{i=1}^{n} p_{t,i} (\ell_i(x_t) - 1)^2.$$

Bounding $(\ell_i(x_t) - 1)^2 \leq 1$, the first claim follows. Minimizing with respect to $\alpha_{\mathsf{p}}$ gives the final claim.

## A.7 Proof of Theorem 4

For $k \in [2, \infty)$, we proceed identically as in Section A.3 to obtain

$$\sum_{t=1}^{T} \mathbb{E}[\ell(x_t)^\top (p - p_t)] = \sum_{t=1}^{T} \mathbb{E}[\widehat{\ell}_t(x_t)^\top (p - p_t)] \leq \frac{B_{\psi_{\mathsf{p}}}(p_t, p_1)}{\alpha_{\mathsf{p}}} + \frac{\alpha_{\mathsf{p}}}{2} \sum_{t=1}^{T} \mathbb{E}\left[\left(\sum_{i=1}^{n} p_{t,i}\right)^3 \sum_{i:p_{t,i}>0} p_{t,i}^{1-k}\right]$$

where the extra summation term appeared since $p_t$'s are no longer normalized. We note that $B_{\psi_{\mathsf{p}}}(p_t, p_1) \leq n^{-k} \rho$ since (8) still holds.

From the definition of $C_k = \max\{t : f_k(t) \leq t\} \vee \frac{\rho}{n}$, we have

$$\sum_{i=1}^{n} n p_i \leq \sum_{i:n p_i \leq C_k} n p_i + \sum_{i:n p_i > C_k} f(n p_i) \leq n C_k + \rho \leq 2 n C_k$$

for all $p \in \mathcal{P}_{\rho,n,\delta}$. Hence, it follows that

$$\sum_{t=1}^{T} \mathbb{E}[\widehat{\ell}_t(x_t)^\top (p - p_t)] \leq \frac{n^{-k}\rho}{\alpha_\mathsf{p}} + 8\alpha_\mathsf{p} T C_k^3 \delta^{1-k} n^k.$$

Minimizing with respect to $\alpha_\mathsf{p}$, we obtain the first result.

When $k \in (1, 2]$, we proceed identically and use the fact that $k_* \geq 2$ and $\ell \in [-1, 0]$ in Lemma 3. Plugging this into the bound (6) and taking expectations, we obtain the second claim by following identical steps as in the case $k \geq 2$.

## B   Updates for $p$

In this section, we will explicitly write down the computations required for mirror descent updates in $p \in \mathcal{P}_{\rho,n}$. The updates for $p$ is

$$p_{t+1} := \operatorname*{argmin}_{p \in \mathcal{P}_{\rho,n}} B_{\psi_\mathsf{p}}(p, w_{t+1}) \tag{17}$$

where $w_{t+1} = \nabla \psi_\mathsf{p}(p_t) + \alpha \widehat{\ell}_t(x_t)$. In the following, we omit subscripts for ease of notation. Note that for $k \leq 1$, since $\|\nabla \psi_\mathsf{p}(p)\| \to \infty$ as $p_i \to 0$ for any $1 \leq i \leq n$, we can ignore the nonnegativity constraint in (17).

### B.1   Power divergence for $k \in (-\infty, 1) \setminus \{0\}$

Writing down the Lagrangian for the optimization problem (17) with $\psi_\mathsf{p}(p) = \frac{1}{k(k-1)} \sum_{i=1}^{n} p_i^k$, we have

$$\mathcal{L}(p, \eta, \lambda) = \frac{1}{k(k-1)} \sum_{i=1}^{n} (p_i^k - w_i^k) - \frac{1}{k-1} \sum_{i=1}^{n} w_i^{k-1}(p_i - w_i)$$
$$- \eta(p^\top \mathbb{1} - 1) - n^{-k}\lambda \left( \rho - \frac{1}{k(k-1)} \sum_{i=1}^{n} ((np_i)^k - 1) \right)$$

where $\eta \in \mathbb{R}^n$ and $\lambda \geq 0$. In any case, the first order conditions for $p$ yield

$$(1+\lambda)p^{k-1} = w^{k-1} + (k-1)\eta\mathbb{1}.$$

Plugging this in the constraint $f$-divergence constraint $\sum_{i=1}^{n} p_i^k \leq n^{-k}(k(k-1)\rho + n)$ and using strict complementarity, we have

$$\lambda(\eta) = \left( \left( \frac{n^k}{k(k-1)\rho+n} \right)^{\frac{1}{k_*}} \left\| w^{k-1} + (k-1)\eta\mathbb{1} \right\|_{k_*} - 1 \right)_+ .$$

Plugging this in the Lagrangian

$$\mathcal{L}(\eta) = \min_{\lambda \geq 0} \max_{p \in \mathcal{P}_{\rho,n}} \mathcal{L}(p, \eta, \lambda) = B_{\psi_\mathsf{p}}(p(\eta), w) - \eta(p(\eta)^\top \mathbb{1} - 1)$$

where $p(\eta) = (1+\lambda(\eta))^{1-k_*}(w^{k-1} + (k-1)\eta\mathbb{1})^{k_*-1}$. Now, it remains to minimize $\mathcal{L}(\eta)$. Noting that $\mathcal{L}(\eta)$ is a concave function, the derivative $\frac{d}{d\eta}\mathcal{L}(\eta)$ is an nondecreasing function. Hence, we can run a bisection search to find $\eta$ such that $\frac{d}{d\eta}\eta = 0$. To this end, compute

$$\frac{d}{d\eta}\mathcal{L}(\eta) = (1+\lambda(\eta))^{1-k_*} \left( \frac{\lambda'(\eta)}{k-1} - 1 \right) \sum_{i=1}^{n} (w_i + (k-1)\eta)^{k_*-1}$$
$$- (1+\lambda(\eta))^{2-k_*} \frac{\lambda'(\eta)}{k-1} \sum_{i=1}^{n} w_i^{k-1}(w_i^{k-1} + (k-1)\eta)^{k_*-2}$$
$$- \eta\lambda'(\eta)(1+\lambda(\eta))^{2-k_*} \sum_{i=1}^{n} (w_i^{k-1} + (k-1)\eta)^{k_*-2}$$

where

$$\lambda'(\eta) = \begin{cases} (k-1)\left(\frac{n^k}{k(k-1)\rho+n}\right)^{\frac{1}{k*}} \left\|w^{k-1}+(k-1)\eta\mathbb{1}\right\|_{k*}^{1-k*} \sum_{i=1}^{n}(w_i^{k-1}+(k-1)\eta)^{k*-1} & \text{if } \lambda(\eta) \geq 0 \\ 0 & \text{otherwise.} \end{cases}$$

Since evaluating $\frac{d}{d\eta}\mathcal{L}(\eta)$ takes $O(n)$ time, the bisection on $\eta$ will find a $\epsilon$-accurate solution in $O(n\log\frac{1}{\epsilon})$ time. Using this optimal $\eta$ to to compute $p(\eta)$ takes another $O(n)$ time.

## B.2 KL divergence ($k = 1$)

Lagrangian for the optimization problem (17) with $\psi_{\mathsf{p}}(p) = \sum_{i=1}^{n} p_i \log p_i$ is

$$\mathcal{L}(p,\eta,\lambda) = \sum_{i=1}^{n} p_i \log \frac{p_i}{w_i} - \eta(p^\top\mathbb{1}-1) - \frac{\lambda}{n}\left(\rho - \sum_{i=1}^{n} np_i \log(np_i)\right).$$

The first order conditions for $p$ yield $p_i = w_i^{\frac{1}{1+\lambda}} n^{-\frac{\lambda}{1+\lambda}} \exp\left(\frac{\eta}{1+\lambda}\right)$ and from $p^\top\mathbb{1} = 1$, it follows that $p_i = w_i^{\frac{1}{1+\lambda}}/\sum_{i=1}^{n} w_i^{\frac{1}{1+\lambda}}$. Plugging this back into the Lagrangian, we have

$$\mathcal{L}(\lambda) = \min_\eta \max_{p\in\mathcal{P}_{\rho,n}} \mathcal{L}(p,\lambda,\eta) = \lambda\left(\log n - \frac{\rho}{n}\right) - \alpha\widehat{\ell}_{I_t}(x_t) - (1+\lambda)\log\sum_{i=1}^{n} w_i^{\frac{1}{1+\lambda}}.$$

Taking derivatives, we get

$$\frac{d}{d\lambda}\mathcal{L}(\lambda) = \log n - \frac{\rho}{n} - \log\sum_{i=1}^{n} w_i^{\frac{1}{1+\lambda}} - \frac{\sum_{i=1}^{n} w_i^{-\frac{\lambda}{1+\lambda}}}{\sum_{i=1}^{n} w_i^{\frac{1}{1+\lambda}}}$$

which can be computed in $O(n)$ flops. Since $\mathcal{L}(\lambda)$ is concave, $\lambda \geq 0$ such that $\frac{d}{d\lambda}\mathcal{L}(\lambda) = 0$ can be found to $\epsilon$-accuracy in $O(n\log\frac{1}{\epsilon})$. Then, the update $p(\eta)$ takes $O(n)$ to compute.

## B.3 EL divergence ($k = 0$)

Lagrangian for the optimization problem (17) with $\psi_{\mathsf{p}}(p) = -\sum_{i=1}^{n} \log p_i$ is

$$\mathcal{L}(p,\eta,\lambda) = -\sum_{i=1}^{n} \log \frac{p_i}{w_i} - \eta(p^\top\mathbb{1}-1) - \lambda\left(\rho + \sum_{i=1}^{n} \log(np_i)\right).$$

The first order conditions for $p$ yield $p_i = (1+\lambda)(\frac{1}{w_i} - \eta)^{-1}$. Plugging this into the divergence constraint and using strict complementarity, we have

$$\lambda(\eta) = \left(\exp\left(\frac{1}{n}\sum_{i=1}^{n} \log\left(\frac{1}{nw_i} - \frac{\eta}{n}\right) - \frac{\rho}{n}\right) - 1\right)_+.$$

Then, it suffices to solve

$$\mathcal{L}(\eta) = \min_{\lambda\geq 0} \max_{p\in\mathcal{P}_{\rho,n}} \mathcal{L}(p,\lambda,\eta) = \sum_{i=1}^{n} p_i(\eta)\log\frac{p_i(\eta)}{w_i} - \eta(p(\eta)^\top\mathbb{1}-1).$$

From concavity, we can run bisection search on the monotone function $\frac{d}{d\eta}\mathcal{L}(\eta)$ to find its zero. To this end, compute

$$\frac{d}{d\eta}\mathcal{L}(\eta) = \sum_{i=1}^{n}\left\{p_i'(\eta)\left(\log\frac{p_i(\eta)}{w_i} - \eta - 1\right) + p_i(\eta)\right\} + 1$$

where

$$p_i'(\eta) = (1+\lambda(\eta))\left(\frac{1}{w_i} - \eta\right)^{-2} + \lambda'(\eta)\left(\frac{1}{w_i} - \eta\right)^{-1}$$

$$\lambda'(\eta) = \begin{cases} -\frac{1}{n}\exp\left(\frac{1}{n}\sum_{i=1}^{n}\log\left(\frac{1}{nw_i} - \frac{\eta}{n}\right) - \frac{\rho}{n}\right)\sum_{i=1}^{n}\left(\frac{1}{w_i} - \eta\right)^{-1} & \text{if } \lambda(\eta) > 0 \\ 0 & \text{otherwise.} \end{cases}$$

Hence, the update $p(\eta)$ can be computed in $O(n)$ flops.

## B.4 Power divergences ($k > 1$)

After some calculations, we have that

$$
\begin{aligned}
g'(\lambda) = {} & \frac{\partial}{\partial\lambda} B_{\psi_{\mathsf{p}}}(p(\lambda), w) + \lambda \frac{\partial}{\partial\lambda} \sum_{i=1}^{n} f_k(np_i(\lambda)) \\
= {} & \left\{ \frac{n^{k+1}\lambda}{(k-1)^2} - \frac{n}{k-1} + \lambda(\lambda-1)\frac{n^{2k+1}}{(k-1)^2} \right\} \left(1 + n^k\lambda\right)^{k_* - 1} \sum_{i \in I(\lambda)} (w_i^{k-1} + n\lambda)^{k_* - 1} \\
& + \left\{ \frac{n\lambda}{(k-1)^2} + \lambda(\lambda-1)\frac{n^{k+2}}{(k-1)^2} \right\} \left(1 + n^k\lambda\right)^{k_*} \sum_{i \in I(\lambda)} (w_i^{k-1} + n\lambda)^{k_* - 2} \\
& + \frac{n^k}{k(k-1)} \left(1 + n^k\lambda\right)^{k_*} \sum_{i \in I(\lambda)} w_i^{k-1}(w_i^{k-1} + n\lambda)^{k_*} \\
& + \left\{ \lambda(\lambda-1)\frac{n^{2k+1}}{(k-1)^2} - \frac{n^{2k}\lambda}{(k-1)^2} \right\} \left(1 + n^k\lambda\right)^{k_* - 1} \sum_{i \in I(\lambda)} w_i^{k-1}(w_i^{k-1} + n\lambda)^{k_* - 1} \\
& + \left\{ \lambda(\lambda-1)\frac{n^{k+2}}{(k-1)^2} - \frac{n^{k+1}\lambda}{(k-1)^2} \right\} \left(1 + n^k\lambda\right)^{k_*} \sum_{i \in I(\lambda)} w_i^{k-1}(w_i^{k-1} + n\lambda)^{k_* - 2} \\
& - \frac{\delta^k - k\delta}{k(k-1)}|I(\lambda)| - \rho + \frac{n\lambda}{k} + \frac{n(\delta^k - k\delta)}{k(k-1)}
\end{aligned}
$$

where

$$
I(\lambda) = \left\{ 1 \le i \le n : w_i^{k-1} \ge \left(\frac{\delta}{n}\right)^{k-1} \left(1 + n^k\lambda\right) - \lambda n \right\}.
$$

Hence, we can run bisection search on $\lambda \ge 0$ to find the zero of the monotone function $\frac{\partial}{\partial\lambda}\mathcal{L}(\lambda)$ as before.

When $k = 2$, under the change of variables $\lambda = n^2\lambda$, we have

$$
\begin{aligned}
\frac{\partial}{\partial\lambda}g(\lambda) = {} & \frac{1}{(1+\lambda)^2} \sum_{i \in I(\lambda)} \left(w_i - \frac{1}{n}\right)^2 - \frac{\rho}{n^2} + \frac{(1-\delta)^2}{2n^2}(n - |I(\lambda)|) \\
= {} & \frac{1}{(1+\lambda)^2} \sum_{i \in I(\lambda)} w_i^2 - \frac{2}{n(1+\lambda)^2} \sum_{i \in I(\lambda)} w_i \\
& + \left( \frac{1}{n^2(1+\lambda)^2} - \frac{(1-\delta)^2}{2n^2} \right) |I(\lambda)| + \frac{(1-\delta)^2}{2n} - \frac{\rho}{n^2}
\end{aligned}
\tag{18}
$$

Making the additional change of variables $\alpha = \lambda/(1+\lambda)$ and $I(\alpha) = \left\{ i : (1-\alpha)w_i + \alpha/n \ge \frac{\delta}{n} \right\}$, we have

$$
\begin{aligned}
\frac{\partial}{\partial\alpha}g(\alpha) = {} & \frac{1}{2} \sum_{i \in I(\alpha)} w_i^2 - \frac{1}{n} \sum_{i \in I(\alpha)} w_i + \frac{1}{2n^2(1-\alpha)^2} \left( (1-\alpha)^2 - (1-\delta)^2 \right) |I(\alpha)| \\
& + \frac{1}{2n^2(1-\alpha)^2} (n(1-\delta)^2 - 2\rho),
\end{aligned}
\tag{19}
$$

which is non-increasing in $\alpha \in [0, 1]$.

## C  Procedures for Efficient Updates when $k = 2$

We detail the operations involving the balanced binary search tree (BST) required for Algorithm 1. The weights $w$ are stored up to multiplicative and additive factors *mult* and *addi*. Each node in the

BST stores the following variables:

$$
\begin{aligned}
i =&\ \text{index in } \{1, \dots, n\} \text{ of node} \\
\text{left} =&\ \text{pointer to the left child. } \emptyset \text{ if empty (NULL)} \\
\text{right} =&\ \text{pointer to the right child. } \emptyset \text{ if empty (NULL)} \\
w =&\ \text{weight, stored up to multiplicative and additive factors (\textit{mult} and \textit{addi})} \\
N_l =&\ \text{number of weights in the left subtree (smaller weights)} \\
N_r =&\ \text{number of weights in the right subtree (bigger weights)} \\
S_l =&\ \text{sum of weights in the left subtree (smaller weights)} \\
S_r =&\ \text{sum of weights in the right subtree (bigger weights)} \\
S_l^2 =&\ \text{sum of squared weights in the left subtree (smaller weights)} \\
S_r^2 =&\ \text{sum of squared weights in the right subtree (bigger weights)}
\end{aligned}
$$

By computing $1 + N_l + N_r$ at the root node, the number of elements in the BST is available in constant time.

We first give the pseudo-code for the sampling procedure used in Line 1.3 of Algorithm 1. *Sample(tree)* samples a node from the given tree with probabilities proportional to the weights of the nodes. At any given node, the procedure decides whether to stay at the current node or recurse down the tree by tossing a coin proportional to the current weight $w$ (stay) and the sum of weights $s_l$ (go left) and $s_r$ (go right). The algorithm returns the node if the coin flip results in a "stay" decision or it reaches a leaf node. By virtue of this recursive strategy, the sampling procedure requires $O(\log n)$.

---

**Algorithm 2** Sample $I_t$

---

1: $coin \leftarrow \text{Uniform}(0,1)$
2: node $\leftarrow root$
3: **while** node is not a leaf **do**
4:     **if** $coin < \frac{1}{1 + \text{node}.N_l + \text{node}.N_r}$ **then**
5:         **return** node
6:     **else if** $coin < (1 + \text{node}.N_l)/(1 + \text{node}.N_l + \text{node}.N_r)$ **then**
7:         node $\leftarrow$ node.left
8:     **else**
9:         node $\leftarrow$ node.right
10:     **end if**
11: **end while**
12: **return** node

---

Next, we briefly outline the procedure for updating the sampled node with index $I_t$ from $p_t$ to $w_{t+1}$. Using the standard BST operations `Remove` and `Insert`, this step requires time $O(\log n)$. For example, a red-black tree uses subtree rotations to update and maintain the values $N_l, N_r, S_l, S_r, S_l^2, S_r^2$ along with the weights in logarithmic time [11]. See Duchi et al. [14] for explicitly updates when storing subtree weights and counts, as in our case.

---

**Algorithm 3** Update w

---

1: Input: $p_{t,I_t}, w_{t,I_t}, I_t$
2: `Remove`$(p_{t,I_t}, I_t)$, `Insert`$(w_{t,I_t}, I_t)$
3: **return** $root$

---

We next give a procedure that computes an $\epsilon$-accurate solution to $\frac{\partial}{\partial \alpha} g(\alpha) = 0$ as in expression (19). We first bisect on the nodes to find the node with its weight at the optimal threshold. Then, we bisect on $\alpha$ to compute the exact value. Since the algorithm proceeds in two bisection steps, it only takes $O(\log n + \log \frac{1}{\epsilon})$ time.

---
**Algorithm 4** Compute $\alpha^*$
---

1: node $= root$, node$_r$, node$_l = \emptyset$
2: $c_{num}, c_{sum}, c_{sum^2} = 0, \ell_{num}, \ell_{sum}, \ell_{sum^2} = 0$
3: **while** true **do**
4:     $w \leftarrow \text{node}.w, \quad \alpha \leftarrow (\delta - nw)/(1 - nw)$
5:     $g(\alpha) \leftarrow \frac{1}{2}(c_{sum^2} + w^2 + \text{node}.S_r^2) - \frac{1}{n}(c_{sum} + w + \text{node}.S_r)$
6:         $+ \frac{1}{2n^2(1-\alpha)^2}((1-\alpha)^2 - (1-\delta)^2)(c_{num} + 1 + \text{node}.N_r) + \frac{1}{2n^2(1-\alpha)^2}(n(1-\delta)^2 - 2\rho)$
7:     **if** $g(\alpha) < 0$ **then**    // too small, increase $\alpha$
8:         node$_r \leftarrow$ node
9:         **if** node.right $= \emptyset$ **then** break
10:        **end if**
11:        node $\leftarrow$ node.right
12:     **else**               // too big, decrease $\alpha$
13:        node$_l \leftarrow$ node
14:        $c_{num} \leftarrow c_{num} + 1 + \text{node}.N_r, \quad c_{sum} \leftarrow c_{sum} + \text{node}.w + \text{node}.S_r$
15:        $c_{sum^2} \leftarrow c_{sum^2} + \text{node}.w^2 + \text{node}.S_r^2$
16:        $\ell_{num} \leftarrow c_{num}, \ell_{sum} \leftarrow c_{sum}, \ell_{sum^2} \leftarrow c_{sum^2}$
17:        **if** node.left $= \emptyset$ **then** break
18:        **end if**
19:        node $\leftarrow$ node.left
20:     **end if**
21: **end while**
22: **if** node$_l \neq \emptyset$ **then**
23:     $c_{num} = \ell_{num}, c_{sum} = \ell_{sum}, c_{sum^2} = \ell_{sum^2}$
24: **end if**
25: $u \leftarrow 1, l \leftarrow 0, \alpha \leftarrow .5$
26: **while** $u - l > \epsilon$ **do**
27:     **if** $g(\alpha, \ell) < 0$ **then**
28:         $u \leftarrow \alpha$
29:     **else**
30:         $l \leftarrow \alpha$
31:     **end if**
32: **end while**
33: Update mult $\leftarrow (1-\alpha)$mult,   addi $\leftarrow (1-\alpha) * \text{addi} + \alpha/n$
34: **return** $\alpha$

---

In Line 4.27, we used $g(\alpha, \ell)$ to denote $g(\alpha)$ as computed with $\ell_{num}, \ell_{sum}, \ell_{sum^2}$ as the relevant sums.

Provided $\lambda^* = 1/(1 - \alpha^*)$, Algorithm 5 gives a procedure for updating the tree to $p(\lambda^*)$ in $O(\log n)$ time. By virtue of the updates (11), we have $p_i(\lambda) \geq \frac{\delta}{n}$ for $i \neq I_t$ since $w_i \geq \frac{\delta}{n}$. Hence, the only potential truncation is for index $I_t$, which takes $O(\log n)$ time by removing and reinserting the node into the tree.

---
**Algorithm 5** Update p
---

1: Input: $\lambda^*$, $w_{t,I_t}$, $I_t$
2: **if** $w_{t,I_t} < \frac{\delta}{n}$ **then**
3:     // If modified weight was too low, truncate.
4:     `Remove`($w_{t,I_t}, I_t$), `Insert`($\frac{\delta}{n}, I_t$)
5: **end if**

---