[Reviews · NeurIPS 2016]

Reviewer 1

Summary

The focus of this paper is in solving a distributionally robust formulation of empirical risk minimization: min_x: max_p: \sum_i p_i loss_i(x) where the p_i represent the weight of the losses, and the uncertainty in these weights represents the distributional uncertainty. The focus of this paper is developing an efficient computational approach. The statistical performance of this method is studied in another paper by (presumably) the same authors, though the results here are only briefly described, and are otherwise unavailable in order to preserve anonymity of the submission. The starting point is considering the standard robust optimization dual approach, whereby the inner max is dualized, yielding a (convex) minimization problem. Yet, this does not seem to yield to stochastic gradient procedure, due to failure to control the variance of the sub gradients. Instead, the paper takes a two-player game approach, using work by Nemirovski as a starting point, as well as recent work by Hazan and others. some minor typographical errors line 16. “In tasks, such as…” grammatically incomplete sentence. line 184. efficiently

Qualitative Assessment

It seems important for the reader to understand better what the results are the one obtains with this paper, that differ from applying Nemirovski’s saddle point results directly. Another related paper is that of Ben-Tal et al., “Oracle-based robust optimization via online learning,” as the general approach proposed there seems similar in spirit to the ideas here. As far as I can see, conceptually the same framework is there, namely, of not using the standard dualization technique of robust optimization, but of staying within the max min framework. What does the current approach get us that we could not do with saddle point approaches, or with the Ben-Tal et al. paper?

Confidence in this Review

2-Confident (read it all; understood it all reasonably well)


Reviewer 2

Summary

The paper solved an expansion of robust empirical risk minimization problem using strategy from two-layer game and mirror descent method. Mirror descent is defined using f-divergences whose many particular cases were investigated.

Qualitative Assessment

It looks like that the regret bounds depend on specific expression of f-divergence. Do you have a way to unify these bounds in one theorem?

Confidence in this Review

2-Confident (read it all; understood it all reasonably well)


Reviewer 3

Summary

The paper considers a class of distributionally robust optimization problems and proposes a specialized first-order optimization algorithm for solving these problems as min-max saddle-point problem.

Qualitative Assessment

My primary feedback was listed above as a "fatal flaw," although again, I'd prefer to describe it as a serious concern. As secondary feedback, I'd point out that the literature review in the paper is somewhat misleading at places. For example, the authors attribute the asymptotic result in eq. 2 and the expansion in eq. 3 both to an anonymous work by the same authors as the submission, but results like the one in eq. 2 have been established in earlier works such as Ben-Tal et. al "Robust Solutions of Optimization Problems Affected by Uncertain Probabilities" and Bertsimas et. al "Robust Sample Average Approximation." Similarly, the result in 3 has been established in Lam "Robust sensitivity analysis for stochastic systems." Were this a journal submission, I'd expect a much fairer treatment of the past results in this field in the literature review.

Confidence in this Review

2-Confident (read it all; understood it all reasonably well)


Reviewer 4

Summary

This paper focuses on solving the robust empirical risk minimization problem, which is in a "min-max" structure and has the ability to automatically trade between loss and variance. Previously this problem is usually solved by using SGD after taking the dual of the inner supremum, however it might not work well because of the unbounded variance of the subgradients. This paper presents a new solver for the robust empirical risk minimization problem. It is inspired by a mirror descent based method to solve two-player convex games. The solver consists of two alternating mirror descent/ascent steps, one is for minimization and the other one is for maximization. The main difference between this work and previous works is that the authors generalize this method to Cressie-Read family f-divergence constrained cases. The authors proved regret bounds for different cases of f-divergences in the Cressie-Read family, including important cases like KL divergence and Chi-square divergence. The convergence shown is sublinear, in O(\sqrt T), which match the regret bounds of common stochastic methods. For the special case of Chi-square divergence, the authors uses a trick proposed in [11] to run the projection step in O(log n) time. The authors also conduct experiments on synthetic and real data, and try to show that the proposed solver is better than SGD and IPM (interior point method) in terms of CPU time and convergence, also the robust solution does provided stability as expected.

Qualitative Assessment

Overall I think the work in this paper is valid and valuable. The first 3 sections are well written. However I think the experiment section looks weak and needs more refinements (see below) before the paper can be published. In the experiments, the proposed algorithm uses the best step size given in section 3, but for SGD, Armijo's backtracking line search is used. I think this could be a reason why SGD is slower, because line search is slow. The authors should compare the two algorithms both using line search, or both using a fixed *best* step size. In the introduction section the authors show that stochastic gradient may fail due to the unbounded objective and subgradients. It would be nice to show a dataset that has this problem, and how the two-player bandit mirror descent algorithm can improve that. This will make this paper stronger and more people will pick it up. Some more explanations and discussions should be added for Figure 2a and 2b, and the legends on some figures can be made better (like in figure 1, "SGD-minmax" can be replaced with "proposed" to avoid confusion with the SGD algorithm being compared). Some missing references (shown as ?) should also be fixed. In section 4, the texts do not explicitly say which Bregman divergence is used. Following the derivation (in KKT condition) I think it's $\frac{1}{2} \sum_1^n p_i^{2}$. Maybe you can make it more clear in section 4. Also, does this O(log n) trick for solving \lambda work for other Bregman divergences?

Confidence in this Review

2-Confident (read it all; understood it all reasonably well)


Reviewer 5

Summary

This paper considers robust empirical risk minimization problems, and proposed an efficient algorithm. The main idea is to consider the robust ERM problem as a two player game, and use the mirror descent to solve the problem. Regret bounds and numerical results are provided.

Qualitative Assessment

This paper focus on making reliable decisions under the data-driven framework, which an interesting and important topic. However, this paper is not carefully written. For example, the references are missing on page 6, line 192 and page 7, line 206. The legend of red lines are missing for Figure 2c,d. The paper states only the necessary information but not sufficient for the readers to follow easily. I think the clarity of this paper could be greatly improved especially the authors did not use the full 8 pages. In terms of technical contributions, even though the main idea is novel (to the best of my knowledge), my major concern is that it seems to ignore the line of works which focus on the same issue (making reliable decisions using distributionally robust optimization). See for example the references below. Therefore, my humble opinion is the more justifications (numerical and theoretical/reasoning) are needed in order to claim the contributions in this topic. In the numerical section, I am confused that the authors claim robust solution provides stability (page 7 line 217). For the figure I could not see this point, and ERM clearly outperforms robust solution in terms of both error and loss. @ARTICLE{2015arXiv150505116M, author = {{Mohajerin Esfahani}, P. and {Kuhn}, D.}, title = "{Data-driven Distributionally Robust Optimization Using the Wasserstein Metric: Performance Guarantees and Tractable Reformulations}", journal = {ArXiv e-prints}, archivePrefix = "arXiv", eprint = {1505.05116}, primaryClass = "math.OC", keywords = {Mathematics - Optimization and Control, Statistics - Computation}, year = 2015, month = may, adsurl = {http://adsabs.harvard.edu/abs/2015arXiv150505116M}, adsnote = {Provided by the SAO/NASA Astrophysics Data System} } @ARTICLE{2015arXiv150909259S, author = {{Shafieezadeh-Abadeh}, S. and {Mohajerin Esfahani}, P. and {Kuhn}, D. }, title = "{Distributionally Robust Logistic Regression}", journal = {ArXiv e-prints}, archivePrefix = "arXiv", eprint = {1509.09259}, primaryClass = "math.OC", keywords = {Mathematics - Optimization and Control, Statistics - Machine Learning}, year = 2015, month = sep, adsurl = {http://adsabs.harvard.edu/abs/2015arXiv150909259S}, adsnote = {Provided by the SAO/NASA Astrophysics Data System} } == post-rebuttal update== I have read the authors' rebuttal. They address some of the concerns but my general opinion has not changed: More justifications (numerical and theoretical/reasoning) are needed in order to claim the potential impact of this work.

Confidence in this Review

2-Confident (read it all; understood it all reasonably well)


Reviewer 6

Summary

The robust empirical risk minimization approach has nice properties (the arXiv paper) -- trades off empirical loss and (non-convex) variance; has an uniformly convergent expansion, etc. Present paper obtains regret bounds for an alternating algo they present (the uncertainty sets can be described by many f-divergences, including KL) and interesting implementations (using BST, red-and-black trees, etc.) that are scalable are also presented. Experiments on synthetic as well as real data are presented. Details in appendix seem adequate -- all the five sections look nice and cover various aspects (and reasonably self contained -- proofs of lemma 1, 2 and 3)).

Qualitative Assessment

This is a very good paper, no issue; will have impact. Perhaps, I missed it, is P_0 (line 31) is the unknown underlying distribution? Useful if some more details on the significance of the f-divergences considered (Cressie-Read family) are provided (of course, this family includes KL). Agreed, one way to generate uncertainity sets, one can leverage their tractability, etc., but any additional aspects? Similarly, useful if significance of LHS terms of the 4 regret bounds obtained are presented. As mentioned by them, effort has been put to make each iterate O(log n). Any comment on the overall complexity? At least some comments based on their experience?

Confidence in this Review

2-Confident (read it all; understood it all reasonably well)